# The pesticides carbofuran and picloram alter the diversity and abundance of soil microbial communities

Jaqueline Alves Senabio[1], Rafael Correia da Silva[2‡], Daniel Guariz Pinheiro[3‡], Leonardo Gomes de Vasconcelos[4‡], Marcos Antônio Soares[1]*

1 Department of Botany and Ecology, Institute of Biosciences, Federal University of Mato Grosso, Cuiabá, Mato Grosso, Brazil, 2 Center for Molecular Biology and Genetic Engineering, State University of Campinas, Campinas, SP, Brazil, 3 Faculty of Agricultural and Veterinary Sciences, Paulista State University, Jaboticabal, SP, Brazil, 4 Department of Chemistry, Federal University of Mato Grosso, Cuiabá, Mato Grosso, Brazil

⊙ These authors contributed equally to this work.
‡ RCS, DGP and LGV also contributed equally to this work.
* drmasoares@gmail.com

**Data Availability Statement:** Raw sequence data was deposited on the NCBI SRA database under the accession code PRJNA833723.

## Abstract

Many countries widely use pesticides to increase crop productivity in agriculture. However, their excessive and indiscriminate use contaminates soil and other environments and affects edaphic microbial communities. We aimed to examine how the pesticides carbofuran and picloram affect the structure and functionality of soil microbiota using cultivation-independent methods. Total DNA was extracted from microcosms (treated or not with pesticides) for amplification and metabarcoding sequencing for bacteria (16S gene) and fungi (28S gene) using Illumina—MiSeq platform. Data analysis resulted in 6,772,547 valid reads from the sequencing, including 3,450,815 amplicon sequences from the V3-V4 regions of the 16S gene and 3,321,732 sequences from the 28S gene. A total of 118 archaea, 6,931 bacteria, and 1,673 fungi taxonomic operating units were annotated with 97% identity in 24 soil samples. The most abundant phyla were *Proteobacteria*, *Actinobacteria*, *Acidobacteria*, *Firmicutes*, *Chloroflexi*, *Euryarchaeaota*, and *Ascomycota*. The pesticides reduced the diversity and richness and altered the composition of soil microbial communities and the ecological interactions among them. Picloram exerted the strongest influence. Metabarcoding data analysis from soil microorganisms identified metabolic functions involved in resistance and degradation of contaminants, such as glutathione S-transferase. The results provided evidence that carbofuran and picloram shaped the soil microbial community. Future investigations are required to unravel the mechanisms by which soil microorganisms degrade pesticides.

## Introduction

Fungicides, nematicides, insecticides and herbicides are used to maximize agricultural production and meet the increased demand for food due to human population growth [1–3].

**Funding:** This work was supported by grants from National Council for Scientific and Technological Development (CNPq, grant #409062/2018–9), The Mato Grosso State Research Foundation (FAPEMAT, grant # 568258/2014), Coordination for the Improvement of Higher Education Personnel (CAPES)—Financial Code 001, and National Council for Scientific and Technological Development (CNPQ, grant # 140677/2021-6).

**Competing interests:** The authors have declared that no competing interests exist.

Pesticides are an integral part of world agriculture that are used in both small- and large-scale crops to control weeds and diseases caused by insects, nematodes and fungi. Even when used correctly, their high persistence and penetration into the soil negatively impact ecosystems [1, 4]. Pesticides at high concentrations and their biotransformation products can persist in ecosystems and interact with their abiotic and biotic parts according to their chemical properties, concentration and target [1, 3, 5].

There are different classes of pesticides, such as organophosphates, organochlorines [5, 6], carbamates [1], synthetic pyrethroids [7], and biopesticides [8]. The herbicide picloram (4-amino-3,5,6-trichloro-2-pyridinecarboxylic acid) and insecticide carbofuran (2,3-dihydro-2,2-dimethyl-7-benzofuranyl-*N*-methyl carbamate) are pesticides usually applied in agriculture. Picloram is a broad-spectrum organochlorine herbicide with strong water solubility that is used to control weeds in pastures [2, 9]. Carbofuran is one of the most toxic insecticides and nematicides from the class I N-methyl carbamate pesticides with ester and amide linkage [10] that is widely used in agricultural, domestic and industrial practices due to its broad pest control spectrum [1].

Although essential in modern high-yield agriculture, pesticides continue to pose risk to human health, animals and the environment due to their extensive use and improper handling. Inappropriate use of pesticides can interfere with soil properties, such as fertility, respiration, biomass and microbial diversity, nitrogen and phosphate mineralization capacity, and enzyme activity [1, 5, 11, 12].

Most of the agricultural pesticides do not have specific targets and thereby affect non-target organisms and harm the environment and edaphic microbiota [3]. Pesticides alter the structure (abundance and composition) of the plant microbiome and directly affect the productivity and fitness of the host [13]. Their negative impact on soil microbiota causes loss of microbial diversity and functionality and reduce soil health and fertility [14].

The soil microbiota is essential to maintain biogeochemical processes and the balance and fertility of the soil ecosystem [15, 16]. The soil microorganisms are indicators of productivity and environmental disturbance of this ecosystem [3]. The understanding of the structure and diversity of microbial communities in contaminated soils helps to optimize bioremediation strategies and performance. The microbial community structure responds in a unique way to various biotic and abiotic conditions [17, 18] and the synergistic effects of microbial activities directly or indirectly favor bioremediation processes [12, 19, 20]. Advances in DNA sequencing technology associated with the development of computational methods for metadata analysis have prompted initiatives to study the diversity and activity of microbial communities during degradation and detoxification of contaminated environments [15, 21, 22].

We hypothesize that contamination with the pesticides carbofuran and picloram and exposure time influence the structure and composition of soil microbial communities, by decreasing species richness and values of diversity indices and shaping microbial interaction networks. In this sense, the present study aims to examine the temporal influence of the pesticides carbofuran and picloram on the structure and functionality of soil microbial community under microcosm conditions.

## Material and methods

### Soil collection and microcosm assembly

The soil samples used in the experiments were collected in a wetland region of the Pantanal biome (S 16˚21'19.7" W 056˚20'13.9") (Sisbio collection license: 24237), with an average temperature of 27.69˚C, relative humidity of 72.53%, and average monthly evapotranspiration of 3.85 mm.day$^{-1}$, during the collection period [23]. This soil has no history of previous exposure

to pesticides; hence, its microbial communities have not undergone influence of these chemical agents. Composite samples were obtained from the soil surface (10 cm deep), homogenized, and sieved through a sieve (2 mm mesh). The soil was transported in a refrigerated box and kept cool until use. The physicochemical characteristics of the soil were previously determined [24].

The microcosms were established in twelve sterile 250 mL glass beakers containing 100 g of soil each. Three samples were treated with picloram (64 mg kg $^{-1}$ of soil), three samples were treated with carbofuran (30 mg kg $^{-1}$ of soil), and three samples were used as control (without pesticide). The flasks were sealed with aluminum foil to prevent deposition of particles from the atmosphere and incubated in the dark at 28°C. The humidity of each beaker was adjusted every two days with autoclaved distilled water, using a volume determined by the lost mass weighed on a semi-analytical scale. The soil in the microcosms was homogenized with the aid of a sterilized glass rod before sampling on the 15th and 30th day. Ten grams of soil were collected for total DNA extraction and quantification of pesticides in the samples. The experiment was carried out in triplicate.

## Soil residual pesticide analysis

Three successive extractions were performed to extract soil residual pesticides. Methanol: water (60:40 v/v) acidified with 0.1% $H_3PO_4$ was used to extract picloram, as reported by Marileo [25], and acetonitrile:water (50:50 v/v) acidified with 1% formic acid was used to extract carbofuran, as reported by Ruiz-Hidalgo [26]. The pesticides were quantified in a high-performance liquid chromatography system tander mass spectrometry (HPLC/MS-ESI), equipped with a binary pump system (Varian ProStar, model 210), automatic injection system (Varian ProStar, Model 410), mass detector (Varian, Model Varian 500-MS), and data acquisition and processing software (Varian MS Workstation, Version 6.9.3). Five µL of the samples were injected into a Kromasil 100-5-C18 column (250×4.6 mm, particle size 5 µm; serial number: E66320; stationary phase C18 Reverse Phase, separation mode: Reverse Phase) and Kromasil Guard pre-column 100-5-C18 (4.6×10 mm, particle size 5 µm, stationary phase C18 Reverse Phase, separation mode: Reverse Phase). Both pesticides were eluted with a mixture of solvent A = 0.1% formic acid (v/v) and solvent B = acetonitrile, at a flow rate of 1.0 mL/min, injected at 40°C. Picloram was eluted under a gradient from 0–10 min with 80% solvent B; 10–12 min with 80% to 95% solvent B; 12–22 min with 95% solvent B; 22–24 min with 95% to 80% solvent B; 24–40 min with 80% solvent B. Carbofuran was eluted under a gradient from 0–10 min with 80% solvent B; 10–11 min with 80% to 95% solvent B; 11–21 min with 95% solvent B; 21–22 min with 95% at 80% solvent B; 22–32 min with 80% solvent B.

The mass spectrometer jet ionization source (electrospray) operated under the following conditions: needle voltage: +/- 5800 V; spray shield voltage: +/- 600V; nebulizer gas pressure: 50 psi; drying gas pressure: 25 psi (picloram) and 35 psi (carbofuran); drying gas temperature: 300°C (picloram) and 320°C (carbofuran); scan type: MS/MS; polarity: positive; capillary voltage: 80V; precusor ion (m/z): 242.0 (picloram) and 221.0 (carbofuran); RF loading: 100%, acquisition time: 10 min. The concentration was determined by comparison with a standard curve prepared for each pesticide, at the concentration range 0.25–0.0019 µg/mL, with R2 = 0.994 (picloram) and R2 = 0.998 (carbofuran).

## Total DNA extraction and metabarcoding sequencing

Total DNA was extracted from 250 mg of soil samples from the 24 microcosms using the PowerSoil™ DNA Extraction Kit, according to the manufacturer's protocol (Mobio Laboratories, Carlsbad, CA). The 16S rRNA genes were amplified using the primers S-D-Bact-0341-b-

S-17 (5'-`CCTACGGGNGGCWGCAG`-3') and S-D-Bact-0785-a-A-21 (5'-`GACTACHVGGGTATC`
`TAATCC`-3') that flank the hypervariable regions V3 and V4 of this gene [27]. Primers U1 and
U2 were used to amplify the nRLSU-U (nuclear ribosomal large subunit rDNA) region of the
28S rDNA gene [28]. The bacterial and fungal *metabarcodings* were sequenced on an Illumina
—MiSeq platform ($2 \times 300$ bp), according to the manufacturer's recommendations. Twenty-
four samples were processed for library generation. Raw sequence data was deposited on the
NCBI SRA database under the accession code PRJNA833723.

## Bioinformatics analysis

The *metabarcodings* were processed using the USEARCH v11.0.667 suite applications [29].
Sequencing quality was measured using the USEARCH11 commands "fastx_info" and "fas-
tq_eestats". Next, the low-quality tips and adapters were removed using the "fastx_truncate"
command with the pruning parameters of readings R1 and R2, respectively, 18 and 19 bases,
for data from amplicons V3-V4 16S, and 20 and 10 bases, for 28S amplicon data. The
sequences were truncated using the "-truncIen" parameter with sizes of 380 and 210 for 16S
and ITS, respectively. The reads were quality controlled for expected errors using the "-fas-
tq_filter" command, returning strings with a Phred Q score = 20.00. Sequences underwent
dereplication using the "-fastx_uniques" command and removal of "singletons" sequences
using the "-sortbysize" command. The operating taxonomic units (OTUs) were chosen with
the "-cluster_otus" command based on a similarity threshold of 97%. Possible chimeric
sequences were removed using the UCHIME algorithm [30]. To quantify OTUs per sample,
preprocessed readings were remapped using 99% identity with the "-otutab" command. The
distance tree was calculated with the commands "-calc_distmx" for matrix calculation and
"-cluster_aggd" for clustering and output in Newick format. The OTUs sequences received
taxonomic prediction using the SINTAX program [31] as an annotator, and the RDP Database
[32] as a reference, plus mitochondrial and chloroplast sequences extracted from the Silva
database 132 [33], with a minimum prediction confidence ("-syntax_cutoff") of 90%. The ref-
erence fungal database was the UNITE version 8.0 [34] for the intergenic spacer sequences
(ITS), with a minimum confidence of acceptance of the prediction of 50% for each taxonomic
level. The sequences that were not annotated by the first stage of SINTAX underwent a new
annotation by alignment with blastn of the BLAST+ tool [35], using the NT database (NCBI
RESOURCE COORDINATORS. 2014) as reference, with the acceptance parameters: mini-
mum e-value 1e-10, identity greater than 97%, and minimum coverage of 80%.

## Prediction of functional profiles

The functional profile was investigated via Phylogenetic Investigation of Communities by
Reconstruction of Unobserved States (PICRUSt2) [36]. Matrices with metabolic pathway
(KO) data were processed using Microbiome Analyst software (https://www.
microbiomeanalyst.ca/) [37], based on data from the KEGG database [38]. Data were filtered
keeping data with prevalence in at least 20% of the samples and normalized by the Cumulative
Sum Scaling (CSS) algorithm. KOs were processed using the Bray-Curtis similarity index. The
differential abundance of each KO was analyzed using the edgeR algorithm with an adjusted
p-value of 0.001 by the False Discovery Rate (FDR) method.

## Statistical analysis

Singleton OTUs were removed from the analyzed data. The relative abundance (%) of individ-
ual taxa in each community was estimated by comparing the number of sequences assigned to
a specific taxon versus (*vs*) the total number of sequences obtained for that sample. The non-

parametric Mann-Whitney U test was used to determine how strongly the variables (pesticide —presence *vs*. absence; time—15 days *vs*. 30 days) influenced the relative abundance of the communities analyzed. The described levels of significance were adjusted for FDR and *P* <0.05 (q value) were considered statistically significant.

The Chao-1 richness estimator and diversity indices (Shannon, Simpson, and J evenness), NMDS and ANOSIM were analyzed using PAST v3.25 software [39]. The paired and/or unpaired t test was used to analyze the statistical difference of Chao-1, Shannon, Simpson, and J evenness regarding the variables (pesticide—presence *vs* absence; time—15 days *vs* 30 days) using the R software v3.5.1 [40].

Venn diagrams were constructed on the Bioinformatics & Evolutionary Genomics platform (http://bioinformatics.psb.ugent.be/webtools/Venn/) using microbial OTUs. The indicator microbial OTUs were analyzed using the IndVal function of the labdsv package [41] in the R v3.5.1 software [40], in order to identify the microbial OTUs that significantly correlate between treatments (presence and absence of pesticide; time—15 days and 30 days).

Molecular ecological networks of soil microorganisms were constructed separately. To explore all pairwise OTUs associations, correlation scores (Spearman and Pearson) were calculated using the R software v3.5.1 [40] with the Hmisc [42] and corrplot [43] packages. Only significant correlations ($P$ <0.05) were maintained and the resulting correlation data were imported into the Cytoscape 3.5.1 software [44]. The NetworkAnalyzer tool was used to calculate topological parameters [45] and the ModuLand plug-in with default parameters was used to calculate modules [46].

## Results

### Pesticide residual concentration in soil

Carbofuran and picloram soil concentration decreased by 72.73 and 18.08% (T test, $P$ <0.05), respectively, after 15 days of incubation, and by 93.43 and 94.63% (T test, $P$ <0.05), respectively, after 30 days of incubation. Therefore, pesticides degradation occurred over time, and their residual concentrations are reported in S1 Table in S1 File.

### Pesticides affect alpha and beta diversity of soil microbiota

We obtained 6,772,547 valid reads using the Hiseq platform: 3,450,815 16S reads and 3,321,732 28S reads (S2 Table in S1 File). Trimming resulted in 5,080,886 high-quality reads– 2,190,590 16S reads and 2,890,296 28S –and clustered OTUs with identity threshold > 97% (S2 Table in S1 File). We eliminated singletons, i.e. the OTUs that appeared only once in the complete sample dataset.

Picloram-treated soil samples provided 62 and 3625 OTUs assigned to 6 and 156 archaeal and bacterial genera, respectively (S3 and S4 Tables in S1 File), and 853 OTUs associated with 127 fungal families (S5 Table in S1 File). Carbofuran-treated soil samples provided slightly lower values: 56 (7 genera), 3,306 (133 genera), and 820 (110 families) OTUs for archaea, bacteria and fungi, respectively (S6-S8 Tables in S1 File). The OTUs for archaea, bacteria, and fungi from the analyzed sequences that were not classified at the phylum level represented, respectively, 2.89, 21.30, and 5.07% in carbofuran-treated samples and 0.6, 13.97, and 2.14% in picloram-treated samples (Fig 1).

Considering all classified OTUs, including pesticide and control treatments, two archaeal, 21 bacterial and three fungal phyla were identified in the soil samples (Fig 1). *Elusimicrobia* and *Ignavibacteriae* were exclusively found in carbofuran-treated samples, while *Aminicenantes* was exclusively found in picloram-treated samples (Fig 1). The most abundant phyla in soils not treated with the pesticides were *Euryarchaeaota* (96.65 and 88.89%), *Proteobacteria*

(18.28 and 18.06%), *Actinobacteria* (17.30 and 20.67%), *Acidobacteria* (17.05 and 15.65%), *Firmicutes* (10.29 and 14.52%), *Chloroflexi* (6.53 and 6.09%), and *Ascomycota* (85.98 and 79.16%)–the first and second values in parentheses refer to control samples for carbofuran and picloram, respectively (S1 and S3 Figs). The pesticides altered the relative abundance of soil microbial communities (S1–S4 Figs). Carbofuran significantly lowered the relative abundance of *Chlamydiae* (41.36%) and *Bacteroidetes* (48.07%), markedly increased the relative abundance of phylum *WPS2* (145.23%) (Mann- Whitney U, $P < 0.05$; FDR adjusted q-value < 0.05) (S1B Fig), but did not alter the frequency of archaeal and fungal phyla (S1A and S1C Fig).

Picloram significantly augmented the abundance of *Euryarchaeaota* (11.62%) and *Firmicutes* (231.94%), reduced the abundance of *Thaumarcheaota* (99.09%), *Saccharibacteria* (79.95%), *WPS2* (64. 74%), *Chloroflexi* (53.87%), *Proteobacteria* (49.78%), WPS1 (49.12%), *Acidobacteria* (46.31%), and *Actinobacteria* (45.55%) (Mann-Whitney U Test, $P < 0.05$; adjusted FDR q-value < 0.05) (S3A and S3B Fig), but did not alter the relative abundance of fungal phyla (S3C Fig).

In soil samples treated with carbofuran and picloram, respectively: (i) the most frequent archaeal genera were *Methanothrix* (30.97 and 28.12%), *Methanocella* (25.60 and 35.60%), *Methanosarcina* (16.92 and 13.37%) and *Methanobacterium* (13.48 and 10.40%) (S5A and S6A Figs); (ii) the most frequent bacterial genera were *Gp1* (7.05 and 5.59%), *Mycobacterium* (3.73 and 4.52%), *Sphingomonas* (2.56 and 2, 30%) and *Gp3* (2.40 and 1.76%) (S5B and S6B Figs); and (iii) the most abundant fungal families were *Pleosporaceae* (16.22 and 5.20%), *Trichocomaceae* (12.06 and 15.72%), *Aspergillaceae* (8.53 and 6.83%), *Lindgomycetaceae* (7.83 and 1.95%) and *Lophiostomataceae* (7.14 and 10.44%) (S1C and S3C Figs).

Compared with the control treatment, carbofuran significantly increased the abundance of the bacterial genus *WPS*2 (192.36%), reduced the abundance of *Subdivision*3 (6.32%) (Mann-Whitney U Test, $P < 0.05$; FDR adjusted q-value < 0.05) (S5B Fig), but did not alter the abundance of archaeal genera and fungal family (S5A and S5C Fig).

Picloran increased the abundance of *Methanocella* (21.66%), *Methanothrix* (18.66%), *Methanosarcina* (8.16%), *Terriglobus* (254,723.81%), *Clostridium* (393.37%), *Tumebacillus* (368, 58%) *Bacillus* (304.86%) and *Lysinibacillus* (243.75%) (Mann-Whitney U Test, $P < 0.05$; FDR adjusted q-value < 0.05) (S6A and S6B Fig), as well as increased the abundance of *Clavicipitaceae* (3,225.17%), *Ophiocordycipitaceae* (776.38%), *Aspergillaceae* (282.25%), *Rhynchogastremataceae* (248.95%), *Trichocomaceae* (101.55%), and *Chaentomiaceae* (54.20%) when compared with control samples (Mann-Whitney U test, $P < 0.05$; FDR adjusted q-value < 0.05) (S6C Fig).

The exposure time to pesticides influenced the edaphic microbiota. After 30 days of treatment, carbofuran reduced the abundance of *WPS*2 (7.60%) and *Subdivision*3 (1.92%) (S2B Fig), while picloram decreased the abundance of the archaea *Methanomassiliicoccus* (26.54%) and the fungal family *Cunninghamellaceae* (71.71%) (S4A and S4C Fig), but significantly increased relative abundance of *Coniochaetaceae* (1,441.54) (Mann-Whitney U test, $P < 0.05$; FDR adjusted q-value < 0.05) (S4C Fig).

We calculated alpha diversity descriptors such as richness (Chao-1), diversity (Shannon and Simpson indices) and evenness (J) to compare the pesticides effect on soil over time (S9 Table in S1 File). Neither carbofuran nor the exposure time altered the estimated alpha diversity indices of bacterial and archaeal OTU communities (S9 Table in S1 File) (t-test, $P < 0.05$). This pesticide only slightly increased the Shannon and Simpson diversity indices in the fungal OTUs (S9 Table in S1 File). In contrast, picloram strongly impacted the diversity of bacterial and fungal OTUs (S9 Table in S1 File).

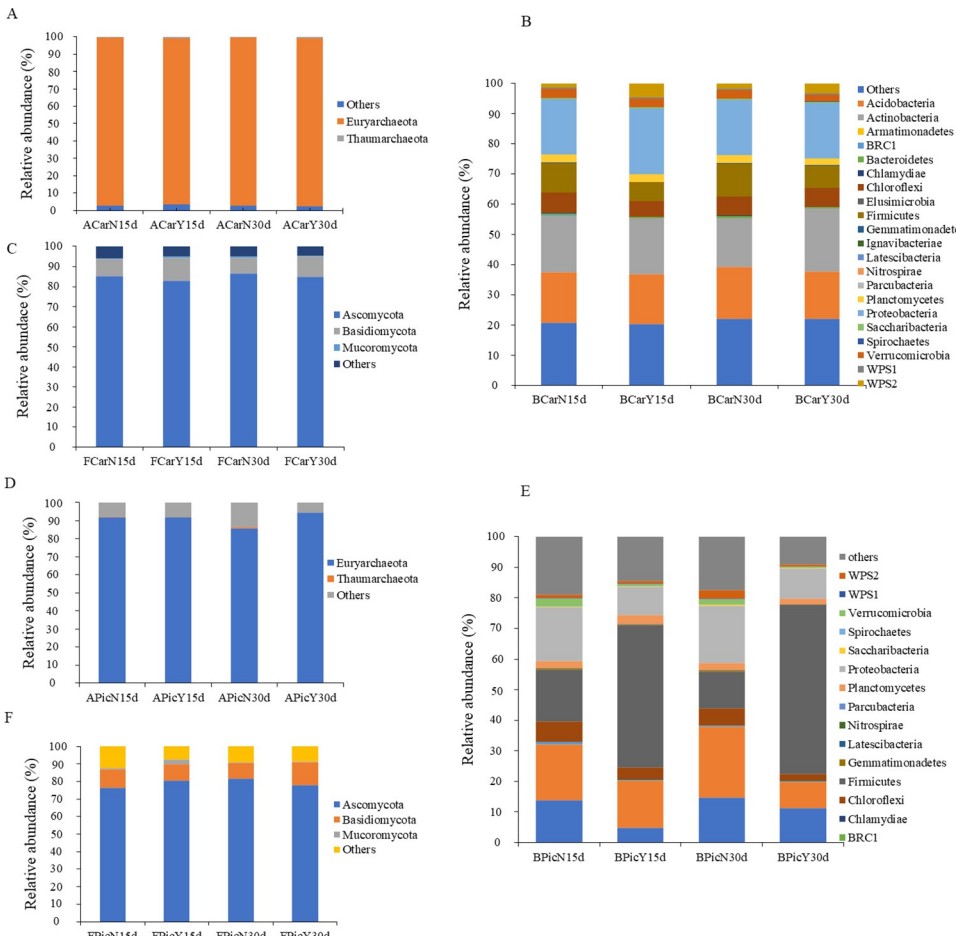

**Fig 1. Composition and relative abundance at phylum level.** (a and d) archaea; (b and e) bacteria (c and f) and fungus, in soils treated (Y) or not (N) with the pesticides carbofuran (Car) and picloram (Pic), respectively, for 15 and 30 days.

To examine how pesticides and exposure time affected the composition and abundance of OTUs in the microbiota, we used the Jaccard and Bray-Curtis similarity indices, respectively (Fig 2 and S7 Fig). Carbofuran only altered the abundance of archaeal and bacterial OTU communities (ANOSIM, $P < 0.05$) (Fig 2 and S10 Table in S1 File). In contrast, picloram and exposure time influenced the composition and abundance of bacterial, archaeal, and fungal communities (ANOSIM, $P < 0.05$) (S7 Fig and S10 Table in S1 File). The treatment periods of 15 and 30 days with carbofuran and picloram shared most archaeal, bacterial and fungal OTUs (S8 and S9 Figs). Compared with shared OTUs, specific OTUs were present in smaller quantities in all treatments (S8 and S9 Figs).

Some taxa that indicated the addition of pesticides to the soil were identified by IndVal analysis (S11 and S12 Tables in S1 File). Carbofuran-treated soil samples had seven indicator fungal OTUs, while picloran-treated samples had 2,263 and 70 indicator archaeal, bacterial and fungal OTUs, respectively (S11 Table in S1 File). There were no genera (bacteria) or families (fungi) that indicated carbofuran presence (S12 Table in S1 File). *Firmicutes* (78.57%) was the most abundant indicator bacterial taxon for picloram presence, followed by *Proteobacteria* (7.14%) and *Ascomycota* (91.67%) (S12 Table in S1 File). Six bacterial genera (*Desulfitobacterium*, *Terriglobus*, *Cellulosilyticum*, *Syntrophomonas*, *Oxobacter* and *Terrisporobacter*) and

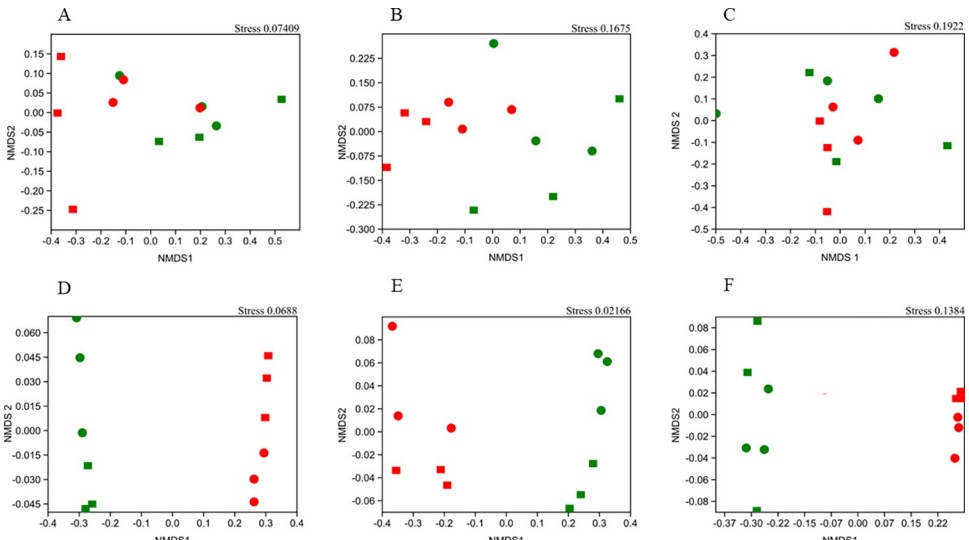

**Fig 2. Non-metric multidimensional scale (NMDS) for Bray-Curtis index of soil microbial communities under microcosm conditions.** (a and d) Archaea; (b and e) Bacteria (c and f) and Fungus, for carbofuran- and picloram-treated soil samples, respectively. Presence and absence of pesticide (green = No, red = Yes) and experimental time (rectangle = 15 days, circle = 30 days).

seven fungal families (*Trimorphomycetaceae*, *Hypocreaceae*, *Clavicipitaceae*, *Chaetosphaeriaceae*, *Ophiostomataceae*, *Herpotrichiellaceae* and *Sporidiobolaceae*) were indicators of picloram presence.

## The pesticides alter molecular ecological network of soil microbial communities in microcosm conditions

We built ecological co-occurrence networks to analyze whether pesticide and exposure time affected the interactions among OTUs. The topological properties of the networks are summarized in S13 Table in S1 File. Carbofuran reduced the number of links for archaea (13.64%), bacteria (2.68%) and fungi (13.07%), while picloram increased the number of links for archaea (12.53%) and fungi (193.10%). Positive interactions predominated in microbial communities from soil treated with both pesticides.

Soil samples treated with the pesticides had less nodes than the untreated samples, except for fungi and bacteria in the picloram-treated samples. Picloram decreased network density only in bacterial communities. Picloram and carbofuran increased the clustering coefficient for archaea and bacteria in soil samples, respectively (S13 Table in S1 File).

## Pesticides alter the microbial community functional profile

The microbiota from carbofuran- and picloram-treated soil samples had 7,065 and 7,104 KO profiles, respectively. The samples treated with carbofuran for distinct periods were similar, indicating that the presence of this pesticide and the exposure time were not sufficient to change the KO abundance profile of the edaphic microbiota. In contrast, the presence of picloram and exposure time resulted in alterations that distinguished the KO abundance profile from the untreated soil samples (S10 Fig).

We identified 5,887 KEGG orthologs in the differentially abundant soil samples treated or not with picloram. We selected the first 25 orthologs for representation according to their lowest FDR values (S14 Table in S1 File), including the proteins glutathione S-transferase

(K00799, EC: 2.5.1.18), gamma-glutamyltranspeptidase/glutathione hydrolase (K00681, EC: 2.3.2.2 /3.4.19.13), 1,3-glucan beta-glucosidase (K01210, EC: 3.2.1.58), N-carbamoyl-D-amino acid hydrolase (K01459, EC: 3.5.1.77), pre-sequence protease (K06972, EC:3.4.24.-), transcriptional regulator of the XRE family, and transcriptional regulator of aerobic/anaerobic benzoate catabolism (K15546). The KO terms from carbofuran data were not significantly different.

## Discussion

Here we demonstrated that the soil levels of carbofuran and picloram decreased to reach final residual concentrations of 6.57% and 5.37% relative to their initial concentrations, respectively. This reduction may be due to the ability of microorganisms to degrade pesticides [47–49]. The bacterial genera *Chryseobacterium* and *Enterobacter* isolated from soil degrade carbofuran *in vitro* [47, 49], while soil fungi and bacteria degrade organochlorine pesticides [50]. Soil microorganisms are one of the largest reservoirs of biodiversity in different ecosystems [51–53] that play key roles in soil ecosystems related to nutrient composition and cycling [54–56]. They have developed resistance and detoxification mechanisms to different pollutants [57–59], which make them objects of interest in bioremediation studies of contaminated soils.

The traditional identification methods are not recommended to perform microbial diversity studies because they detect only a small portion of the real number of microbial communities, which are generally composed of dominant species, and exclude rare species that often play important roles in ecosystems [5, 57, 60]. Methods based on high-throughput sequencing technology have provided more complete data on microbial communities that inhabit different environments [15, 60, 61]. The present study used cultivation-independent identification methods to determine the composition, structure and metabolic profile of soil microbial communities under microcosm conditions. The soil samples analyzed contained archaeal, bacterial and fungal OTUs, with greater richness of bacterial OTUs. Our findings corroborate other studies that report higher richness of bacterial OTUs when compared with fungal and archaeal OTUs [22, 62]. All levels had a percentage of unclassified OTUs and no taxonomic level was completely identified in our study, probably due to the lack of homologous sequences in the databases [63] or insufficient sequencing depth [64].

The predominant phyla in the soil samples were *Acidobacteria*, *Actinobacteria*, *Firmicutes*, *Proteobacteria*, *Euryarchaeaota* and *Ascomycota* (Fig 1), which are also dominant in soil microbial communities [57, 65, 66]. Their key roles in the soil functionality remain known. *Actinobacteria* recycle substances in the environment, including catabolism of complex polymers and production of bioactive compounds or metabolites [67], while *Proteobacteria* mediate nitrogen fixation [68] and plant growth via the carbon cycle, and promote plant growth [69, 70]. The dominance of microbial phyla and their sensitivity to contamination may vary according to the pollutant type. The phylum *Ascomycota* predominates in soils contaminated with metribuzin (herbicide), imidacloprid (insecticide), and benomyl (fungicide) [66], while the phylum *Basidiomycota* predominates in soils contaminated with dibutyl phthalate [71].

The application and residual concentration of herbicides and pesticides alter the structure of the rhizospheric microbiota by increasing or decreasing richness and abundance, as well as by modifying the composition of microbial species [13]. Picloram lowered the relative abundance of *Actinobacteria*, *Acidobacteria*, and *Proteobacteria* (S3 Fig). Carbofuran at the dose applied to the soil samples did not significantly changed microbial abundance. The pesticide dose applied determines its impact on the soil microbiota. Benomyl, metribuzin and imidacloprid at low doses do not clearly change the microbial community [72]. Chlorantraniliprole does not significantly affect bacterial and fungal diversity but changes the structure of bacterial and fungal communities in soil [73].

*Actinobacteria* is sensitive to metal contamination, in contrast to *Acidobacteria* and *Proteobacteria* [16]. Propiconazole reduces the relative abundance of Acidobacteria [74]. Thus, some phyla are more sensitive to certain pollutants. Picloram significantly increased the relative abundance of the phylum *Firmicutes* (S3 Fig). Contamination can enrich soil with microbial species that tolerate high levels of pollutants because they develop structures to resist or adapt to toxic environments. *Firmicutes* form endospores, which is a differentiated resistant structure produced under abiotic stress [75].

The presence of pesticides in the soil is a selective pressure to develop microorganisms that degrade these contaminants; however, it reduces beneficial microorganisms and thereby decrease soil health and fertility [76, 77]. The presence of carbofuran and picloram and the experimental time influenced on the relative abundance of soil microbial communities at the genus level (Fig 2 and S5 Fig). The pesticides increased the relative abundance of the genera *WPS*2, *Methanocella*, *Methanothrix*, *Methanosarcina*, *Methanobacterium*, *Bacillus*, *Lysinibacillus* and *Clostridium* and of the fungal families *Trichocomaceae* and *Aspergillaceae*. Species belonging to these genera and families are well-known for their ability to resist and degrade different toxic chemical compounds [57, 78]. Bacillus sp. efficiently degrade carbofuran [79]. It was probably one of the genera that mediated carbofuran degradation in the soil samples studied herein.

To characterize the response of soil microbial communities to abiotic stress induced by carbofuran and picloram and exposure time, we analyzed alpha (S9 Table in S1 File) and beta (Fig 2 and S7 Fig) diversity from metabarcoding data. Although carbofuran is a highly toxic pesticide [1], it did not significantly affect the richness, diversity and composition of the microbial communities of the soil samples analyzed. However, picloram strongly influenced the indices of diversity, richness and composition of soil microbial communities. Pesticides are one of the main stressors of soil microorganisms with reciprocal effects on ecosystem functioning [1]. Various pesticides, including propiconazole [17], chlorpyrifos, isoproturon, tebuconazole [3, 80], picloram [81], and carbofuran [82] change the richness, diversity and composition of soil microbial communities.

This study identified indicator OTUs for carbofuran- and picloram-treated soil (S11 Table in S1 File). The seven indicator OTUs identified for carbofuran-treated samples belonged to the fungi domain. The 335 indicator OTUs identified for picloram-treated samples were distributed among the three domains, in which *Firmicutes*, *Proteobacteria* and *Ascomycota* were the most frequent indicators. Species from these phyla are more abundant in soil microbial communities [57, 65, 66] and have mechanisms of resistance and degradation to different pollutants [58, 59, 83, 84].

Picloram-treated soil samples had high clustering coefficients for archaeal and fungal OTUs, but low values for bacterial OTUs (S13 Table in S1 File). The clustering coefficient and network density reflect changes in soil microorganism associations and indicate how closely nodes are embedded in their neighborhood and clustered [85]. Network links (correlations) represent potentially synergistic and antagonistic relationships between microbial communities. Density indicates that soil microorganisms form tight associations in response to contaminants [52], such as picloram.

We built co-occurrence networks to analyze the inter-relationship of the soil microbiome, considering that pesticides can affect the ecological interactions among soil microbial OTUs (S13 Table in S1 File). In general, soil samples treated with the pesticides had high number of positive relationships, suggesting that microorganisms cooperated and adapted better to variations caused by pesticides. Similarly, soil microbiota exposed to azoxystrobin and oxytetracycline present more complex interaction networks [86].

Pollutant contamination usually reduces microbial diversity and increases tolerant and resistant species [5, 66]. Alterations in soil microbiota diversity and composition can decline multiple ecosystem functions, including plant diversity, cycling, and nutrient retention [87–89]. Pesticides induce stress on the microbiota, which alters a variety of functional pathways such as those related to degradation of organic compounds and reduction of ABC-type transporters [86]. The main functional characteristics of soil microbiota in the picloram-treated samples identified after analysis of the abundance of KEGG orthologs are listed in S14 Table in S1 File. The microbial communities from carbofuran-treated soil samples had no significant KEGG orthologs profiles, according to the PICRUSt method. Microorganisms from all life domains carry out biochemical transformations that are directly linked to the ecosystem functioning [90]. They play roles related to the production of primary and secondary metabolites for their development or for surrounding organisms, or even tolerate and/or degrade toxic compounds [21, 59, 84, 91]. Carbon degradation metabolic pathways are upregulated in propiconazole-contaminated soils and correlate with its detoxification [74]. For instance, glutathione S-transferase plays important roles in detoxification of physiological compounds, harmful xenobiotics and metals [92, 93], as well as confers resistance to *Rhizoctonia solani* to the fungicide SYP-14288 [94]. Adaptation of microorganisms to stressful environments also requires biological factors. The physiological functions of β-1,3-glucanases mainly depend on their source of origin. In bacteria, they play nutritional role in the hydrolysis of β-1,3-glucans; in fungi, they play physiological roles in morphogenetic and morphological processes during the fungi development and differentiation, and play a key role in interspecific competition by degrading β-1,3-glucans from the cell walls of competitors [16]. Transcription factors of xenobiotic response elements comprise the second family of regulators that occur more frequently in bacteria. As most regulators of the xenobiotic response elements family are hypothetical proteins, it is not possible to predict the general functions of most members of this protein family [95].

## Conclusions

We examined how the pesticides carbofuran and picloram and experimental time influenced on the structure and composition and functional profile of soil microbial communities using metabarcoding data. The presence of pesticides and the experimental time affected soil microbial communities by altering the relative abundance, richness, diversity, and composition of microbial OTUs, as well as by shaping the interaction networks among OTUs. Picloram affected the edaphic microbial community more strongly than carbofuran. We predicted the functional profile and evidenced adaptive responses of microbial communities to picloram using metabarcoding sequencing. Additional studies to explore the different mechanisms by which microorganisms lower the soil picloram concentration shall help to develop new strategies for bioremediation of contaminated areas.

## Supporting information

**S1 Fig. Relative abundance at phylum level in soil samples treated with carbofuran (Car), under microcosm conditions.** (A) Archaea; (B) Bacteria; (C) Fungi. *FDR adjusted q-value <0.05 (Pesticide—presence (CarY) vs. absence (CarN); Mann-Whitney U test). (TIF)

**S2 Fig. Relative abundance at phylum level in soil samples treated with carbofuran (Car) for distinct experimental times, under microcosm conditions.** (A) Archaea; (B) Bacteria; (C) Fungi. * FDR adjusted q-value <0.05 (Time—15 days (Car15d) vs. 30 days (Car30d);

Mann-Whitney U test).
(TIF)

**S3 Fig. Relative abundance at phylum level in soil samples treated with picloram (Pic), under microcosm conditions.** (A) Archaea; (B) Bacteria; (C) Fungi. *FDR adjusted q-value <0.05 (Pesticide—presence (PicY) vs. absence (PicN); Mann-Whitney U test).
(TIF)

**S4 Fig. Relative abundance at phylum level in soil samples treated with picloram (Pic) for distinct experimental times, under microcosm conditions.** (A) Archaea; (B) Bacteria; (C) Fungi. *FDR adjusted q value <0.05 (Time—15 days (Pic15d) vs. 30 days (Pic30d); Mann-Whitney U test).
(TIF)

**S5 Fig. Relative abundance at genus (Archaea and bacteria) and family (Fungi) levels in soil samples treated with carbofuran (Car) and picloran (Pic) at distinct experimental times, under microcosm conditions.** (A and D) Archaea; (B and E) Bacteria; (C and F) Fungi. *FDR adjusted q-value <0.05 (Time—15 days (Car15d; Pic15d) vs. 30 days (Car30d; Pic30d); Mann-Whitney U test).
(TIF)

**S6 Fig. Relative abundance at genus (Archaea and bacteria) and family (Fungi) levels in soil samples treated with carbofuram (Car) and picloram (Pic), under microcosm conditions.** (A and D) Archaea; (B and E) Bacteria; (C and F) Fungi. * FDR adjusted q-value < 0.05 (Pesticide—presence (CarY; PicY) vs. absence (CarN; PicN); Mann-Whitney U test).
(TIF)

**S7 Fig. Non-metric multidimensional scale (NMDS) expressed as Jaccard index of soil microbial communities treated with carbofuran and picloram, under microcosm conditions.** (A and D) Archaea; (B and E) Bacteria; (C and F) Fungi. The pesticide presence and absence are represented in red and green, respectively. The experimental times of 15 and 30 days are represented as rectangles and circles, respectively.
(TIF)

**S8 Fig. Venn diagram of OTUs from microbial communities from soil samples treated with carbofuran (Car) and picloram (Pic), under microcosm conditions.** (A and D) Archaea; (B and E) Bacteria; (C and F) Fungi. Analysis of pesticide presence (CarY; PicY) *vs.* absence (CarN; PicN).
(TIF)

**S9 Fig. Venn diagram of soil microbial communities after treatment with carbofuran (Car) and picloram (Pic) for distinct experimental times, under microcosm conditions.** (A and D) Archaea; (B and E) Bacteria; (C and F) Fungi. Analysis of experimental time of 15 days (Car15d; Pic15d) *vs.* 30d (Car30d; Pic30d).
(TIF)

**S10 Fig. Dendrogram from cluster analysis of KO abundance profile data from soil samples treated with pesticides, under microcosm conditions.** (A) Carbofuran; (B) Picloram.
(TIF)

**S1 File.  S1 Table**. Residual concentrations of carbofuran (Car) and picloram (Pic) in soil samples at 15 and 30 days of incubation under microcosm conditions. **S2 Table.** Total number of sequences of soil samples at different sequencing phases. **S3 Table.** Absolute and relative

abundance of archaeal OTUs in soil samples treated (Y) or not (N) with picloram (Pic) under microcosm conditions, at experimental times of 15 and 30 days. **S4 Table.** Absolute and relative abundance of bacterial OTUs in soil samples treated (Y) or not (N) with picloram (Pic) under microcosm conditions, at experimental times of 15 and 30 days. **S5 Table.** Absolute and relative abundance of fungal OTUs in soil samples treated (Y) or not (N) with picloram (Pic) under microcosm conditions, at experimental times of 15 and 30 days. **S6 Table.** Absolute and relative abundance of archaeal OTUs in soil samples treated (Y) or not (N) with carbofuran (Car) under microcosm conditions, at experimental times of 15 and 30 days. **S7 Table.** Absolute and relative abundance of bacterial OTUs in soil samples treated (Y) or not (N) with carbofuran (Car) under microcosm condition, at experimental times of 15 and 30 days. **S8 Table.** Absolute and relative abundance of fungal OTUs in soil samples treated (Y) or not (N) with carbofuran (Car) under microcosm conditions, at experimental times of 15 and 30 days. **S9 Table.** OTUs diversity in soil samples treated (Y) or not (N) with carbofuran (Car) and picloram (Pic), at experimental times of 15 and 30 days. **S10 Table.** ANOSIM statistical analysis of soil microbial communities in soil samples treated with carbofuran (Car) and picloram (Pic), under microcosm conditions. **S11 Table.** Indicator OTUs of soil microbial communities in soil samples treated (Y) or not (N) carbofuran (Car) and picloram (Pic) under microcosm conditions, at experimental times of 15 and 30 days. **S12 Table.** Indicator taxa of soil microbial communities in soil samples treated (Y) or not (N) with carbofuran (Car) and picloram (Pic) under microcosm conditions, at experimental times of 15 and 30 days. **S13 Table.** Topological parameters of ecological networks from soil microbial communities in soil samples treated (Y) or not (N) with carbofuran (Car) and picloram (Pic) under microcosm conditions, at experimental times of 15 and 30 days. **S14 Table.** Functional prediction summary of soil microorganisms from picloram-treated soil samples, under microcosm conditions.
(XLSX)

**S1 Graphical abstract.**
(TIFF)

## Author Contributions

**Conceptualization:** Jaqueline Alves Senabio, Marcos Antônio Soares.

**Formal analysis:** Jaqueline Alves Senabio, Rafael Correia da Silva, Daniel Guariz Pinheiro, Leonardo Gomes de Vasconcelos, Marcos Antônio Soares.

**Funding acquisition:** Marcos Antônio Soares.

**Investigation:** Jaqueline Alves Senabio, Marcos Antônio Soares.

**Methodology:** Jaqueline Alves Senabio.

**Project administration:** Marcos Antônio Soares.

**Supervision:** Marcos Antônio Soares.

**Writing – original draft:** Jaqueline Alves Senabio, Marcos Antônio Soares.

**Writing – review & editing:** Jaqueline Alves Senabio, Marcos Antônio Soares.

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
