## [Decision Letter · Decision Letter 0]

8 Aug 2024

PONE-D-24-20226The pesticides carbofuran and picloram alter the diversity and abundance of soil microbial communitiesPLOS ONE

Dear Dr. Soares,

Thank you for submitting your manuscript to PLOS ONE. After careful consideration, we feel that it has merit but does not fully meet PLOS ONE’s publication criteria as it currently stands. Therefore, we invite you to submit a revised version of the manuscript that addresses the points raised during the review process.

We look forward to receiving your revised manuscript.

Kind regards,

Luigimaria Borruso

Academic Editor

PLOS ONE

2. In your Methods section, please provide additional information regarding the permits you obtained for the work. Please ensure you have included the full name of the authority that approved the field site access and, if no permits were required, a brief statement explaining why

“This work was supported by grants from National Council for Scientific and Technological Development (CNPq, grant #409062/2018–9), The Mato Grosso State Research Foundation (FAPEMAT, grant # 568258/2014), Coordination for the Improvement of Higher Education Personnel (CAPES)—Financial Code 001, and National Council for Scientific and Technological Development (CNPQ, grant # 140677/2021-6).”

5. In this instance it seems there may be acceptable restrictions in place that prevent the public sharing of your minimal data. However, in line with our goal of ensuring long-term data availability to all interested researchers, PLOS’ Data Policy states that authors cannot be the sole named individuals responsible for ensuring data access (http://journals.plos.org/plosone/s/data-availability#loc-acceptable-data-sharing-methods).

6. Please amend either the abstract on the online submission form (via Edit Submission) or the abstract in the manuscript so that they are identical.

Reviewers' comments:

Reviewer's Responses to Questions

**Comments to the Author**

1. Is the manuscript technically sound, and do the data support the conclusions?

Reviewer #1: Yes

Reviewer #2: Yes

2. Has the statistical analysis been performed appropriately and rigorously? 

Reviewer #1: Yes

Reviewer #2: Yes

3. Have the authors made all data underlying the findings in their manuscript fully available?

Reviewer #1: Yes

Reviewer #2: Yes

4. Is the manuscript presented in an intelligible fashion and written in standard English?

Reviewer #1: Yes

Reviewer #2: Yes

5. Review Comments to the Author

Reviewer #1: The vertical axis is not found in Figure 1. All figures have this problem.

Line 98-99: mg kg-1

Line 194-198: Why is there no significance analysis in the pesticide analysis section?

Line 214-269: The results are divided into too many paragraphs and should be consolidated.

Line 259, 264, 266: The format of the "P" value should be uniform, generally uppercase plus italics, with specific reference to journal requirements.

Line 256-273: I suggest that results of diversity should be moved before the results of community composition.

Line 308: The logical structure of the discussion sections needs further combing, and they have too many paragraphs and need to be properly consolidated and reorganized.

Line: 402-412: The results section needs to be divided into a separate section, and the summary of the results is too shallow and does not highlight the main point. It is well known that the presence of pesticides can change soil microbes, and how to improve them is the focus. Using the metabolome or others?

Line 406-408: This part of the content can be deleted.

Fig S1: The error bar in Figure S1 B?

The font and format of all figures need to be uniform.

I think the full text of the language needs to be further polished by professionals.

Reviewer #2: The research manuscript PONE-D-24-20226 have been submitted to Plos one with title “The pesticides carbofuran and picloram alter the diversity and abundance of soil microbial communities”. The authors investigated the effect of the stress response to the pesticides carbofuran and picloram on soil microbial diversity using DNA metabarcoding. The effect was observed in 24 soil samples, where a total of 118 archaeal, 6,931 bacterial, and 1,673 fungal OTUs were annotated with 97% identity. The study found that soil diversity and function are affected by the presence of pesticides and experiment duration. In my opinion, though the manuscript contains well-spelled hypotheses and makes use of standard procedures,. I, however, recommend accepting the article with major revisions, as some points need to be addressed before this manuscript is suitable for final publication.

The authors need to:

1. Check the similarity index of the manuscript. The percentage similarity is high to be published in the journal (44% with references, 22% without references). I will suggest the affected part should be reworded.

2. Grammatical presentation should be reviewed throughout the manuscript.

3. The authors did not follow the journal format

Specific comments

i. Abstract contains no methodology

ii. I will suggest the introduction be rewritten in such a way that it will flow logically by consolidating the scattered information about the negative impacts of pesticides which should be merged to form a paragraph.

iii. There are insufficient literatures to back your claim on the role of pesticides on the structure and diversity on the soil microbiota.

iv. At what time/period were the soil samples collected at the region? The depth and diameter the soils were taken should be indicated.

v. Also, the temperature and the cumulative rainfall should also be stated.

6. PLOS authors have the option to publish the peer review history of their article (what does this mean?). If published, this will include your full peer review and any attached files.

Reviewer #1: No

Reviewer #2: **Yes: **Olumayowa Mary Olowe

---

## [Author Response · Author response to Decision Letter 0]

4 Oct 2024

Reviewer #1

ANSWER: We appreciate your dedication in reading the article and providing feedback to the editor. Your suggestions were important for improving the manuscript.

The vertical axis is not found in Figure 1. All figures have this problem.

ANSWER: We corrected Figure 1. We insert the vertical axis. Figure 2 has the vertical axis (NMDS2).

Line 98-99: mg kg-1

ANSWER: We corrected it: mg kg -1

Line 194-198: Why is there no significance analysis in the pesticide analysis section?

ANSWER: We will enter information about the statistical test: 

Line 214-269: The results are divided into too many paragraphs and should be consolidated.

ANSWER: We consolidate the paragraphs.

Line 259, 264, 266: The format of the "P" value should be uniform, generally uppercase plus italics, with specific reference to journal requirements.

ANSWER: We change "p" to "P" throughout the text.

Line 256-273: I suggest that results of diversity should be moved before the results of community composition.

ANSWER: Dear reviewer, we would like to count on your understanding in maintaining the alpha diversity results after describing the community composition.

Alpha diversity estimators take into account the composition and abundance of species, and we understand that the reader needs to know the composition to then glimpse the alpha diversity indicators. Furthermore, the alpha diversity results are strategically placed before the beta diversity results (analyzed by the Jaccard and Bray-Curtis similarity indices). We believe that the results have more cohesion and are more didactic in that order.

Line 308: The logical structure of the discussion sections needs further combing, and they have too many paragraphs and need to be properly consolidated and reorganized.

ANSWER: We consolidate and restructure the paragraphs.

Line: 402-412: The results section needs to be divided into a separate section, and the summary of the results is too shallow and does not highlight the main point. It is well known that the presence of pesticides can change soil microbes, and how to improve them is the focus. Using the metabolome or others?

ANSWER: The results are divided into 4 sections to facilitate readers' understanding. We do not use metabolome methods because our hypothesis and objectives are not focused on analyzing microbial community structure. We used the PICRUSt tool to analyze the impact of pesticides on microbiota functionality. This tool allows inference of the functional profile of a microbial community based on marker gene survey along one or more samples..

Line 406-408: This part of the content can be deleted.

ANSWER: We have deleted it. 

Fig S1: The error bar in Figure S1 B?

ANSWER: We have corrected it. 

The font and format of all figures need to be uniform.

ANSWER: We have corrected it. 

I think the full text of the language needs to be further polished by professionals.

ANSWER: We send the manuscript to a professional specialized in English language proofreading.

Reviewer #2:

The research manuscript PONE-D-24-20226 have been submitted to Plos one with title “The pesticides carbofuran and picloram alter the diversity and abundance of soil microbial communities”. The authors investigated the effect of the stress response to the pesticides carbofuran and picloram on soil microbial diversity using DNA metabarcoding. The effect was observed in 24 soil samples, where a total of 118 archaeal, 6,931 bacterial, and 1,673 fungal OTUs were annotated with 97% identity. The study found that soil diversity and function are affected by the presence of pesticides and experiment duration. In my opinion, though the manuscript contains well-spelled hypotheses and makes use of standard procedures,. I, however, recommend accepting the article with major revisions, as some points need to be addressed before this manuscript is suitable for final publication.

ANSWER: The authors thank you for your dedication in reading and pointing out improvements in the quality of the manuscript.

The authors need to:

Check the similarity index of the manuscript. The percentage similarity is high to be published in the journal (44% with references, 22% without references). I will suggest the affected part should be reworded.

Grammatical presentation should be reviewed throughout the manuscript.

The authors did not follow the journal format

ANSWER: We made the necessary adjustments. 

We reviewed the text and used two tools to check for plagiarism. The tools are:

https://plagiarismdetector.net/

https://www.plagium.com/

The results indicate that there is no plagiarism in the article.

Specific comments

i. Abstract contains no methodology

ANSWER: We insert general information about the methodology.

ii. I will suggest the introduction be rewritten in such a way that it will flow logically by consolidating the scattered information about the negative impacts of pesticides which should be merged to form a paragraph.

ANSWER: We rewrote the impact of pesticides on soil microbiota, including other references.

iii. There are insufficient literatures to back your claim on the role of pesticides on the structure and diversity on the soil microbiota.

ANSWER: We included other references in the discussion that assessed the impact of other pesticides on soil microbiota. We kept the references already included in the discussion that support the role of pesticides in the structure of the microbial community

iv. At what time/period were the soil samples collected at the region? The depth and diameter the soils were taken should be indicated.

ANSWER: We include this information in the material and methods. “Composite samples were obtained from the soil surface (10 cm deep), homogenized, and sieved through a sieve (2 mm mesh).”

v. Also, the temperature and the cumulative rainfall should also be stated.

ANSWER: We include this information in the material and methods.

---

## [Decision Letter · Decision Letter 1]

12 Nov 2024

The pesticides carbofuran and picloram alter the diversity and abundance of soil microbial communities

PONE-D-24-20226R1

Dear Dr. Soares,

We’re pleased to inform you that your manuscript has been judged scientifically suitable for publication and will be formally accepted for publication once it meets all outstanding technical requirements.

Kind regards,

Luigimaria Borruso

Academic Editor

PLOS ONE

Additional Editor Comments (optional):

Reviewers' comments:

Reviewer's Responses to Questions

**Comments to the Author**

1. If the authors have adequately addressed your comments raised in a previous round of review and you feel that this manuscript is now acceptable for publication, you may indicate that here to bypass the “Comments to the Author” section, enter your conflict of interest statement in the “Confidential to Editor” section, and submit your "Accept" recommendation.

Reviewer #1: All comments have been addressed

Reviewer #2: All comments have been addressed

2. Is the manuscript technically sound, and do the data support the conclusions?

Reviewer #1: Yes

Reviewer #2: Yes

3. Has the statistical analysis been performed appropriately and rigorously? 

Reviewer #1: Yes

Reviewer #2: Yes

4. Have the authors made all data underlying the findings in their manuscript fully available?

Reviewer #1: Yes

Reviewer #2: Yes

5. Is the manuscript presented in an intelligible fashion and written in standard English?

Reviewer #1: Yes

Reviewer #2: Yes

6. Review Comments to the Author

Reviewer #1: The authors have made some changes according to the comments made earlier.

Line 203: “are” should be replaced with “were”.

Reviewer #2: Dear Authors,

Thank you for taking the time to attend to these corrections. At this point, I believe we have reached a stage where no additional changes will be made from your side. Therefore, the manuscript should be accepted without further modification. All the best.

7. PLOS authors have the option to publish the peer review history of their article (what does this mean?). If published, this will include your full peer review and any attached files.

Reviewer #1: No

Reviewer #2: **Yes: **Olumayowa Mary Olowe

---

## [Editor Report · Acceptance letter]

15 Nov 2024

PONE-D-24-20226R1 

PLOS ONE

Dear Dr. Soares, 

I'm pleased to inform you that your manuscript has been deemed suitable for publication in PLOS ONE. Congratulations! Your manuscript is now being handed over to our production team.

Kind regards, 

on behalf of

Dr. Luigimaria Borruso 

Academic Editor

PLOS ONE